# The influence of mental state attributions on trust in large language models
Clara Colombatto [1] ✉, Jonathan Birch [2] & Stephen M. Fleming [3]

Rapid advances in artificial intelligence (AI) have led users to believe that systems such as large language models (LLMs) have mental states, including the capacity for 'experience' (e.g., emotions and consciousness). These folk-psychological attributions often diverge from expert opinion and are distinct from attributions of 'intelligence' (e.g., reasoning, planning), and yet may affect trust in AI systems. While past work provides some support for a link between anthropomorphism and trust, the impact of attributions of consciousness and other aspects of mentality on user trust remains unclear. We explored this in a preregistered experiment (N = 410) in which participants rated the capacity of an LLM to exhibit consciousness and a variety of other mental states. They then completed a decision-making task where they could revise their choices based on the advice of an LLM. Bayesian analyses revealed strong evidence against a positive correlation between attributions of consciousness and advice-taking; indeed, a dimension of mental states related to experience showed a negative relationship with advice-taking, while attributions of intelligence were strongly correlated with advice acceptance. These findings highlight how users' attitudes and behaviours are shaped by sophisticated intuitions about the capacities of LLMs—with different aspects of mental state attribution predicting people's trust in these systems.

Artificial intelligence (AI) holds great promise for aiding humans in a wide range of tasks, from everyday activities such as personal assistance to more complex tasks such as logistical planning or medical decision-making. While sophisticated abilities enhance the efficiency and accuracy of artificial systems, they also foster in users the impression that these systems possess mental states such as thoughts, emotions, intentions, and even consciousness, the subjective awareness of oneself and the environment. The tendency to assign mental states to AI systems is independent from whether these systems truly possess them—and indeed, the extent to which current AI systems possess consciousness remains contentious in scientific analysis[1–5]. On the contrary, recent surveys have revealed that the majority of a representative sample of the public attributes some possibility of human-like consciousness to large language models (LLMs)[6], and exposure to as few as three answers from these systems leads to an increase in attributions of mental capacities[7]. Irrespective of whether or not attributions of mental states to LLMs cohere with scientific findings, it is thus important to understand how such attributions drive changes in people's usage of these systems.

Here we investigate how users' attributions of mental states to AI impact their trust in its advice. Attributions of mental states to AI can be beneficial for users, insofar as they foster greater understanding, engagement, and motivation. For example, learning in educational contexts is enhanced by anthropomorphic features such as face-like appearance[8], and ratings of human-likeness in AI are correlated with more positive impressions of the systems and overall social health[9]. These positive effects of anthropomorphism on engagement and evaluations may result in increased reliance on AI advice and functions[10]. Past work has provided some support for this notion: anthropomorphism is positively correlated with trust in robots solving arithmetic problems[11], in chatbots on e-commerce platforms[12], and in AI solutions to an image classification task[13]. Anthropomorphism is also associated with greater trust resilience, as people are more resistant to updating their trust in anthropomorphic agents when seeing them err[14]. These effects can have important consequences: in the context of autonomous driving, people trust vehicles more when they display anthropomorphic cues (e.g., name, gender, voice), even when these superficial cues are unrelated to the underlying abilities of the vehicle[15–18].

While extant work provides some support for the association between anthropomorphism and trust, work in social psychology has demonstrated that mental state attribution is not a unitary concept, but rather involves distinct and independent dimensions of 'intelligence' and 'experience'. This

[1]Department of Psychology, University of Waterloo, Waterloo, ON, Canada. [2]Department of Philosophy, Logic and Scientific Method, and Centre for Philosophy of Natural and Social Science, London School of Economics and Political Science, London, UK. [3]Department of Experimental Psychology and Max Planck UCL Centre for Computational Psychiatry and Ageing Research, University College London, London, UK. ✉e-mail: clara.colombatto@uwaterloo.ca

dissociation is supported by factor analyses of ratings of a variety of mental states, which reveal two fundamental factors: agency, or the capacity for high-level reasoning and planning, and experience, or the capacity for sensations and emotions[19] (for a review, see ref. [20]). These dimensions jointly contribute to our evaluations of various agents, from animals to babies, but their dissociation is especially evident in judgments of AI systems. An LLM may be judged to have high levels of intelligence as per its sophisticated language abilities, while also being judged to have little or no capacity to feel emotions[6].

While previous work has suggested that users place greater trust in anthropomorphic systems, manipulations of anthropomorphic appearance impact mental state attributions in several ways, leaving multiple possible explanations for their effect on trust. In other words, a dissociation between intelligence and experience raises the possibility that trust may depend more on users' beliefs about some capacities (e.g., intelligence) than others (e.g., emotions). Here, we sought to explicitly address how attributions of intelligence and experience relate to trust. We recruited a stratified sample of US adults and probed their intuitions about the capacity for consciousness and a variety of other mental states in a prominent LLM, ChatGPT. Participants also completed a general knowledge task, where they made a series of decisions regarding the population of various countries while also receiving advice. Participants were told that the advice was generated by ChatGPT, although the advice was in fact predetermined based on the performance of previous participants, allowing full control over the accuracy and format of the advice. This design allowed us to investigate how attributions of various mental states relate to advice acceptance, supporting broader inferences about how mental state attribution can support or undermine reliance and trust in human-AI interactions.

## Methods

All methods and analyses were preregistered and can be accessed at https://aspredicted.org/fqtk-n7sf.pdf and https://aspredicted.org/bw5k-gpb3.pdf. Anonymized raw data and analysis code are openly available on the Open Science Framework (OSF) website at https://osf.io/w537f/. All methods and procedures were reviewed and approved by a University of Waterloo Research Ethics Board (Protocol #46224), and all participants provided informed consent before taking part in the study.

### Participants

Participants were recruited in July and August 2024 through Prolific Academic (Prolific.com)[21] in exchange for monetary compensation. To be eligible to take part in the study, participants were required to be fluent in English, have an approval rate of 95–100%, and be located in the US. Recruitment was stratified by age and gender categories based on US census data, to obtain a representative sample.

Prior to starting data collection, we determined and preregistered a total sample of 200 participants (post-exclusions; https://aspredicted.org/fqtk-n7sf.pdf, submitted on 07/01/2024). This sample was chosen based on an a priori power analysis, which revealed that a sample of 193 participants would be sufficient to achieve 80% power to detect a weak to moderate correlation ($r = 0.20$) with an alpha level of 0.05. Analyses of this initial sample revealed a weak correlation between consciousness attributions and advice-taking rates, and follow-up Bayesian analyses revealed anecdotal evidence against the null (BF = 0.51). In light of these inconclusive results, we planned and preregistered the recruitment of additional subjects in batches of $N = 50$ (pre-exclusions) until the BF was conclusive (i.e., either <0.3 or >3), or until a maximum of $N = 500$ participants had been acquired (https://aspredicted.org/bw5k-gpb3.pdf, submitted on 07/25/2024). Analyses of these additional participants continued to show inconclusive results, and so we reached our maximum sample size of $N = 500$. Given that the decision to increase the sample size (from 200 to a maximum of 500) was taken following the analysis of the first 200 participants, and given the reduced evidentiary value of frequentist probabilities in sequential sampling methodologies, we report below both frequentist and Bayesian analyses in the final sample. Bayesian analyses were conducted in JASP[22] using default

priors (i.e., a Beta(1,1) prior for the correlation coefficient; a Cauchy distribution with a scale of $r = 0.707$ for $t$-tests; a Jeffreys–Zellner–Siow (JZS) prior with a scale of 0.354 for linear regressions; and a generalized g-prior distribution (CCH) with alpha = 0.5, beta = 2, $s = 0$ for logistic regressions)[23] and default Markov chain Monte Carlo settings (i.e., Bayesian adaptive sampling (BAS) with 1000 samples for linear and logistic regressions).

Participants were excluded if they (1) reported having encountered problems (as assessed via a question in debriefing; $N = 7$); (2) failed to answer any of our open-ended questions sensibly ($N = 1$); and (3) failed to answer correctly at least 70% of the catch questions in the advice-taking task ($N = 82$). No participants triggered the additional exclusion criteria of participating multiple times or having a browser window smaller than $720 \times 500$ px. A total of 410 participants were thus included in the analyses (204 women, 199 men and 7 others; mean age = 47.19, all self-reported; data on race or ethnicity were not collected).

### Procedure

Participants were redirected to a website where stimulus presentation and data collection were controlled via custom software written in HTML, CSS, and JavaScript, using the jsPsych library[24] and the JATOS server[25]. After participants provided their informed consent, their browser window was put in full-screen mode. Each participant completed two tasks, in a randomized order (with 209 participants completing the advice-taking task first, and 201 participants completing the mental state ratings first). Each task was preceded by instructions specific to the current task, and the first also included a general introduction to ChatGPT: "ChatGPT is an artificial intelligence chatbot developed by OpenAI and released in November 2022. The name 'ChatGPT' combines 'Chat', referring to its chatbot functionality, and 'GPT', which stands for generative pre-trained transformer, a type of large language model (LLM). ChatGPT gained attention for its detailed and articulate responses spanning various domains of knowledge. By January 2023, it had become the fastest growing consumer software application in history, gaining over 100 million users".

**Advice-taking task**. Participants made a series of choices in a general knowledge task, and had the opportunity to update their answers after receiving advice (Fig. 1A). The advice was generated based on the choices of previous participants, as further described in the following paragraph. This allowed us to control the accuracy and format of the advice provided to participants. However, participants were told that the advice they would receive was generated by ChatGPT. Each trial started with a question about which of the two countries had a smaller or larger population (e.g., "Which country has a larger population: Colombia or Germany?"). After participants responded, they were provided with advice, and they were given the opportunity to revise their initial choice (e.g., "Your choice: Colombia. ChatGPT's choice: Germany. Would you like to change your initial choice? Press 'y' for yes, 'n' for no."). On some trials (with 10% probability), they were asked to report which option ChatGPT had selected on the last trial, followed by feedback ("Correct!" or "Wrong") for 500 ms. Each participant completed a total of 40 trials.

The choices of the LLM were determined by matching each participant with one of 10 counterpart participants who had previously completed the general knowledge task[26]. For these counterpart participants, we first selected the 53 countries with populations between 25 and 500 million based on the 2021 UN census (https://population.un.org/wpp/), and randomly selected pairs of countries out of all possible combinations and asked participants to select the one with the smaller or larger population—resulting in 176 trials (2 questions ["smaller", "larger" framing] × 2 country positions [smaller, larger on the left] × 44 country pairs). In the current experiment, each participant was assigned to view 40 randomly selected trials from a randomly selected counterpart participant.

**Mental state ratings**. Participants were told that they would be asked about their "opinions about ChatGPT". First, they were asked to rate ChatGPT's capacity for phenomenal consciousness, on a scale from 1

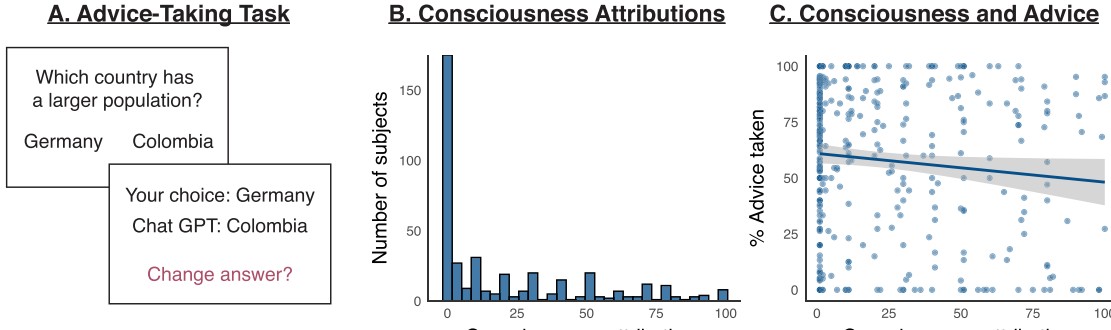

**Fig. 1 | Relationship between consciousness attributions and advice-taking. A** In the advice-taking task, participants ($N = 410$) answered general knowledge questions and had the opportunity to revise their choices after receiving advice from ChatGPT. **B** While 43% of participants denied that ChatGPT could be conscious, the majority (57%) attributed some possibility for consciousness. **C** Bayesian analyses showed no evidence for a significant relationship between consciousness attributions and advice-taking, with strong evidence against a positive correlation—suggesting that participants who attributed higher levels of consciousness were not more willing to take advice from ChatGPT. Error bands represent 95% confidence intervals.

('clearly does not have conscious experiences') to 100 ('clearly has conscious experiences'[6]). Next, they were shown a series of 20 mental states, chosen from prior work[6] to represent dimensions of 'intelligence' (attention, choosing, communication, considering, deciding, intelligence, knowing, memory, planning, and reasoning) or 'experience' (admiration, enjoyment, fear, guilt, happiness, hope, passion, pleasure, relief, and resentment). Participants saw one mental state at a time, in a randomized order, and they were asked to rate the extent to which they thought ChatGPT was capable of each mental state (e.g., "To what extent do you think ChatGPT is capable of planning?"), on a scale from 1 ('not at all') to 100 ('very much'[6]).

At the end of the experiment, participants were asked to report their trust in the LLM on the current task ("How much do you trust ChatGPT's advice on the countries task?") and in general ("How much do you trust ChatGPT's advice in general?"), to be answered on a scale from 'No trust at all' to 'Trust a lot'. They were also asked about their usage frequency of digital assistants ("How often do you use digital assistants, such as Siri (Apple), Google Assistant (Google), or Alexa (Amazon)?"), social AI ("How often do you use any social AI applications such as Replika, Character AI, Talkie, Digi or any other such examples?"), and chatbots ("How often do you use AI chatbots, such as ChatGPT, Claude, Gemini or Grok?"), with response options "More than once a day", "About once a day", "About once a week", "About once every two weeks", "About once a month", "Less than once a month", "I only tried these a couple of times", "I've never used these". They then completed a series of demographic questions and after reading a debriefing letter outlining the purposes of the study, they were redirected to Prolific for remuneration.

## Results

Overall, participants performed the general knowledge task above chance, with an average performance of 62.70% for choices made prior to receiving the advice. On 38.57% of trials, they received advice that conflicted with their initial answer, and they reversed their initial choices on 58.30% of these—resulting in a final accuracy of 63.08%. Accuracy thus did not improve significantly from initial to final choices (exploratory, two-tailed paired test: $t(409) = -0.84$, $p = 0.404$, mean difference $= -0.00$, CI $= [-0.01, 0.01]$, Cohen's $d = 0.04$, $BF_{10} = 0.08$, error $= 0.28\%$), likely because the advice was drawn from previous participants and was thus often incorrect (mean advice accuracy $= 62.48\%$). Advice-taking reflected general trust attitudes towards AI systems, as in exploratory analyses we found that rates of advice-taking were positively correlated with responses on the debriefing questions regarding trust in the LLM both on the current task ($r(408) = 0.58$, $p < 0.001$, 95% confidence interval [CI] $= [0.51, 0.64]$, $BF_{10} = 1.38 \times 10^{+35}$) and in general ($r(408) = 0.41$, $p < 0.001$, CI $= [0.32, 0.49]$, $BF_{10} = 7.52 \times 10^{+14}$).

## Consciousness attributions

As shown in Fig. 1B, attributions of consciousness showed a positive skew, with a considerable proportion of participants selecting no consciousness at all ($N = 175$; 42.68%) but the majority indicating some possibility of phenomenal consciousness (mean [$M$] $= 21.51$, median $= 6$, standard deviation [SD] $= 27.71$, range $= 1-100$). The proportion of participants who indicated ChatGPT has no consciousness at all was higher than in a previous report of data collected 13 months prior to the current sample (33%, reported in ref. [6]). Since this difference was confounded with the addition of the advice-taking task in the current experimental design compared to the earlier survey, in an exploratory analysis we examined a potential effect of task order—but attributions of consciousness were greater in participants who completed the advice-taking task first ($M = 24.18$, SD $= 29.35$) compared to those who completed the ratings first ($M = 18.71$, SD $= 25.67$, Welch two-tailed $t(404) = -2.01$, $p = 0.045$, CI $= [-10.83, -0.12]$, Cohen's $d = 0.20$), and a Bayesian analysis indicated no evidence for a significant difference ($BF_{10} = 0.76$, error $= 0.03\%$).

Our primary preregistered analysis focused on the relationship between advice-taking and consciousness attributions, which is depicted in Fig. 1C. There was a small negative correlation between advice-taking rates and consciousness attributions (Pearson's $r(407) = -0.10$, $p = 0.043$, CI $= [-0.20, -0.00]$; Spearman's rho $= -0.07$, $p = 0.141$). A Bayesian correlation analysis indicated no evidence for a positive or negative correlation compared to a null correlation ($r = -0.10$, $BF_{10} = 0.47$, CI $= [-0.19, -0.00]$), and strong evidence against a positive correlation ($BF_{+0} = 0.02$, CI $= [0.00, 0.06]$).

Given the high positive skew of consciousness attributions in our sample, in an additional exploratory analysis we computed the correlations excluding subjects who rated consciousness as 1 ('clearly does not have conscious experiences'), revealing a numerically stronger yet still weak negative relationship (Pearson's $r(232) = -0.14$, $p = 0.028$, CI $= [-0.27, -0.02]$; Spearman's rho $= -0.14$, $p = 0.027$) with no evidence in Bayesian analyses ($BF_{10} = 0.90$). A preregistered test for the fit of a second-degree polynomial confirmed that the linear predictor was a better fit than a quadratic one ($B = -71.93$, SE $= 35.52$, $t(406) = -2.03$, $p = 0.044$, CI $= [-141.75, -2.11]$; vs. $B = -12.20$, SE $= 35.52$, $t(406) = -0.34$, $p = 0.731$, CI $= [-82.02, 57.62]$). Similar results were obtained with an exploratory generalized linear model with a logistic link function predicting trial-wise choices to obtain advice (no $= 0$, yes $= 1$) from consciousness ratings (rescaled from 0 to 1) with random slopes for subjects ($B = -1.11$, SE $= 0.50$, $z = -2.21$, $p = 0.027$, CI $= [-2.14, -0.12]$, $BF_M = 2.87 \times 10^{+6}$). Contrary to advice-taking behaviour, impressions of trust were positively correlated with consciousness attributions, with a weak correlation with trust on the current task ($r(407) = 0.11$, $p = 0.023$, CI $= [0.02, 0.21]$) which was not significant in

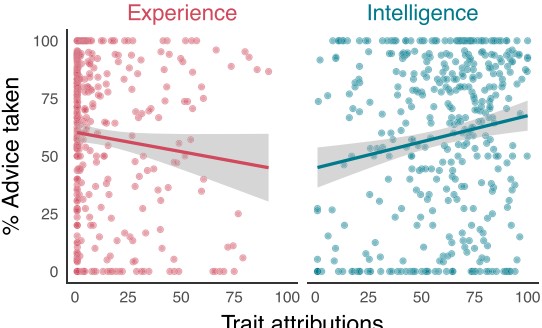

**Fig. 2 | Attributions of mental states to ChatGPT. A** Attributions of mental states clustered around two main dimensions—"experience" (red) and "intelligence" (teal). **B** Rates of advice-taking weakly decreased with higher attributions of experience, while they strongly increased with attributions of intelligence. Each point represents one participant (total N = 410), and error bands represent 95% confidence intervals.

Bayesian analyses ($BF_{10}$ = 0.83) and a strong correlation with trust in general ($r$(407) = 0.25, $p$ < 0.001, CI = [0.16, 0.34], $BF_{10}$ = 43,808.62).

## Mental state attributions

We next investigated the relationship between advice-taking and the attribution of other mental states beyond consciousness. First, we confirmed that the 20 mental states mapped onto the intended dimensions: participants' ratings were reduced via a principal component analysis to two main dimensions explaining 45.30% and 18.79% of the variance, respectively (Fig. 2A). The first mapped onto the previously identified dimension of 'experience', with high loadings of traits such as happiness, enjoyment, passion, hope, and relief. The second instead represented 'intelligence', with high loadings of traits such as memory, deciding, communication, intelligence, and knowing. Overall, attributions were higher for traits related to intelligence ($M$ = 59.39, SD = 24.99) compared to experience ($M$ = 12.25, SD = 19.06; exploratory, two-tailed paired test: $t$(409) = −39.28, $p$ < 0.001, mean difference = −47.14, CI = [−49.50, −44.78], Cohen's $d$ = 1.94, $BF_{10}$ = 9.03 × $10^{+136}$, error = 4.70 × $10^{-139}$%). Preregistered analyses revealed that consciousness attributions were related to experience ($B$ = 1.48, SE = 0.29, $t$(405) = 5.13, $p$ < 0.001, CI = [0.91, 2.05]) and less strongly to intelligence ($B$ = 0.16, SE = 0.04, $t$(405) = 3.65, $p$ < 0.001, CI = [0.08, 0.25]) with a trending interaction between these two factors ($B$ = −0.01, SE = 0.00, $t$(405) = −1.93, $p$ = 0.054, CI = [−0.01, 0.00]). These results were confirmed with Bayesian analyses, which revealed that both experience and intelligence were related to consciousness attributions (experience: $P$(incl) = 0.56, $P$(incl|data) = 1.00, $BF_{inclusion}$ = 2.80 × $10^{+44}$; intelligence: $P$(incl) = 0.56, $P$(incl|data) = 0.96, $BF_{inclusion}$ = 18.39), with anecdotal evidence for their interaction ($P$(incl) = 0.33, $P$(incl|data) = 0.56, $BF_{inclusion}$ = 2.58). Consciousness attributions were best explained by a model including experience, intelligence, and their interaction ($BF_M$ = 2.58), with anecdotal evidence from Bayesian analyses indicating this model had a better fit than models including experience and intelligence but not their interaction ($BF_{10}$ = 2.11), experience only ($BF_{10}$ = 0.22), and intelligence only ($BF_{10}$ = 1.52 × $10^{-44}$).

The relationship between mental state attributions and advice-taking is depicted in Fig. 2B. A preregistered analysis confirmed that the rates of advice acceptance were strongly related to attributions of intelligence ($B$ = 0.31, SE = 0.08, $t$(406) = 3.92, $p$ < 0.001, CI = [0.15, 0.46]), while they were negatively related to experience ($B$ = −1.04, SE = 0.51, $t$(406) = −2.04, $p$ = 0.042, CI = [−2.04, −0.04]), with no significant interaction between these two factors (B = 0.01, SE = 0.01, $t$(406) = 1.37, $p$ = 0.171, CI = [0.00, 0.02]). These results were confirmed with Bayesian analyses, which revealed that both experience and intelligence were related to advice-taking (experience: $P$(incl) = 0.56, $P$(incl|data) = 0.99, $BF_{inclusion}$ = 108.29; intelligence: $P$(incl) = 0.56, $P$(incl|data) = 1.00, $BF_{inclusion}$ = 635.49), with anecdotal evidence for their interaction ($P$(incl) = 0.33, $P$(incl|data) = 0.57,

$BF_{inclusion}$ = 2.61). Advice-taking rates were best explained by a model including experience, intelligence, and their interaction ($BF_M$ = 2.61), with anecdotal evidence from Bayesian analyses indicating this model had a better fit than models including experience and intelligence but not their interaction ($BF_{10}$ = 2.26), intelligence only ($BF_{10}$ = 0.03), and experience only ($BF_{10}$ = 0.00).

Similar results were obtained with an exploratory generalized linear model predicting trial-wise choices to obtain advice with random slopes for subjects (main effect of intelligence: B = 0.44, SE = 0.04, $z$ = 12.47, $p$ < 0.001, CI = [0.37, 0.51]; main effect of experience: $B$ = −0.45, SE = 0.05, $z$ = −9.53, $p$ < 0.001, CI = [−0.54, −0.36]; interaction: $B$ = 0.19, SE = 0.04, $z$ = 4.31, $p$ < 0.001, CI = [0.10, 0.28]). Bayesian analyses of trial-wise choices confirmed a key role for mental state attributions in advice-taking, with attributions of experience-related traits being negatively related to advice-taking and intelligence-related traits being positively related to advice-taking (experience: $P$(incl) = 0.56, $P$(incl|data) = 1.00, $BF_{inclusion}$ = 3.98 × $10^{+24}$; intelligence: $P$(incl) = 0.56, $P$(incl|data) = 1.00, $BF_{inclusion}$ = 1.64 × $10^{+35}$; interaction ($P$(incl) = 0.33, $P$(incl|data) = 1.00, $BF_{inclusion}$ = 7403.05). Indeed, the best fitting model included experience, intelligence, and their interaction ($BF_M$ = 7403.05), which had a better fit than models including experience and intelligence but not their interaction ($BF_{10}$ = 8.11 × $10^{-4}$), intelligence only ($BF_{10}$ = 6.04 × $10^{-25}$), and experience only ($BF_{10}$ = 1.47 × $10^{-35}$).

In an additional exploratory analysis, we examined how the advice-taking task may have impacted attributions of intelligence and experience. We considered that participants' beliefs about the capabilities of ChatGPT may have been influenced by exposure to advice, which was not always accurate but rather comparable to human performance. Attributions of intelligence were similar for participants who completed the advice-taking task first ($M$ = 60.28, SD = 24.38) compared to those who completed the ratings first ($M$ = 58.46, SD = 25.63, Welch two-tailed $t$(405) = −0.74, $p$ = 0.463, CI = [−6.68, 3.04], Cohen's $d$ = 0.07; $BF_{10}$ = 0.14, error = 0.12%). Attributions of experience were also similar for participants who completed the advice-taking task first ($M$ = 13.55, SD = 20.21) compared to those who completed the ratings first ($M$ = 10.89, SD = 17.74, Welch two-tailed $t$(405) = −1.42, $p$ = 0.157, CI = [−6.35, 1.03], Cohen's $d$ = 0.14; $BF_{10}$ = 0.29, error = 0.06%). The mean accuracy of the advice was unrelated to attributions of intelligence ($r$(408) = 0.05, $p$ = 0.325, CI = [−0.05, 0.14], $BF_{10}$ = 0.10) and experience ($r$(408) = 0.03, $p$ = 0.591, CI = [−0.07, 0.12], $BF_{10}$ = 0.07) even when only considering participants who completed these ratings after the advice-taking task (intelligence: $r$(207) = 0.06, $p$ = 0.365, CI = [−0.07, 0.20], $BF_{10}$ = 0.13; experience $r$(207) = 0.07, $p$ = 0.334, CI = [−0.07, 0.20], $BF_{10}$ = 0.14).

The limited impact of exposure to incorrect advice was also clear in exploratory analyses of advice-taking rates. Participants who received more accurate advice on average were not more willing to accept the advice

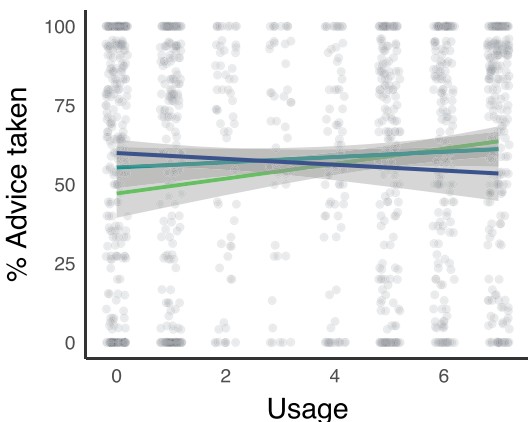

**Fig. 3 | Usage of various AI applications. A** Attributions of consciousness to LLMs were positively related to usage of social AI, but not digital assistants or chatbots. **B** Advice-taking was unrelated to chatbot usage, but was negatively correlated with usage of social AI, and positively correlated with usage of digital assistants. Each point represents one participant (total $N = 410$), and error bands represent 95% confidence intervals.

($r(408) = 0.06$, $p = 0.192$, CI = [−0.03, 0.16], $BF_{10} = 0.14$). Despite exposure to incorrect advice, advice-taking rates did not decrease from the first 20 trials ($M = 56.90$, SD = 36.87) to the last 20 ($M = 59.63$, SD = 38.56, two-tailed paired $t(409) = −2.21$, $p = 0.028$, mean difference = −2.73, CI = [−5.17, −0.30], Cohen's $d = 0.11$; $BF_{10} = 0.62$, error = 0.04%).

## Usage

In an additional set of exploratory analyses, we asked how the main variables of interest, advice-taking and consciousness attributions, were related to usage frequency. In general, participants reported relatively high usage of digital assistants (median = 6 or "once a day", $M = 4.75$, SD = 2.36, range = 0–7) and chatbots (median = 5 or "once a week", $M = 3.91$, SD = 2.43, range = 0–7), while social AI was used less frequently (median = 0 or "never used", $M = 1.68$, SD = 2.31, range = 0–7). As depicted in Fig. 3A, consciousness attributions to ChatGPT were positively related to usage of social AI ($B = 4.50$, SE = 0.64, $t(399) = 7.01$, $p < 0.001$, CI = [3.24, 5.76]), while there was no statistically significant relationship with usage of digital assistants ($B = −0.02$, SE = 0.64, $t(399) = −0.03$, $p = 0.980$, CI = [−1.27, 1.24]) or chatbots ($B = 0.30$, SE = 0.67, $t(399) = 0.45$, $p = 0.656$, CI = [−1.01, 1.60]). Social AI emerged as most related to consciousness attributions also in Bayesian analyses ($P(incl) = 0.50$, $P(incl|data) = 1.00$, $BF_{inclusion} = 1.09 \times 10^{+10}$, $BF_M = 37.04$) compared to digital assistants ($P(incl) = 0.50$, $P(incl|data) = 0.13$, $BF_{inclusion} = 0.15$, $BF_{10} = 7.44 \times 10^{−13}$) and chatbots ($P(incl) = 0.50$, $P(incl|data) = 0.14$, $BF_{inclusion} = 0.16$, $BF_{10} = 9.72 \times 10^{−11}$). On the other hand, as shown in Fig. 3B, advice-taking was negatively correlated with usage of social AI ($B = −2.33$, SE = 0.87, t(400) = −2.66, $p = 0.008$, CI = [−4.05, −0.61]), and positively correlated with usage of digital assistants ($B = 2.98$, SE = 0.87, $p < 0.001$, CI = [1.27, 4.69])—but there was no statistically significant relationship with usage of chatbots ($B = 0.65$, SE = 0.91, $t(400) = 0.71$, $p = 0.476$, CI = [−1.13, 2.43]). These patterns were confirmed in Bayesian analyses, where advice-taking was most related to usage of social AI ($P(incl) = 0.50$, $P(incl|data) = 0.84$, $BF_{inclusion} = 5.12$) and digital assistants ($P(incl) = 0.50$, $P(incl|data) = 0.98$, $BF_{inclusion} = 51.03$), but not chatbots ($P(incl) = 0.50$, $P(incl|data) = 0.37$, $BF_{inclusion} = 0.59$). Indeed, the model including social AI and digital assistants best explained the data ($BF_M = 10.40$), and adding chatbots did not improve the fit ($BF_{10} = 0.24$).

## Discussion

We found a nuanced relationship between mental state attributions and trust: participants more often attributed to an LLM traits related to intelligence (e.g., memory) than those related to experience (e.g., happiness), and attributions of intelligence were strongly related to advice acceptance. Meanwhile, the dimension of mental states related to experience showed a weakly negative relationship with advice-taking.

What might explain the strong association between intelligence-related attributions and advice acceptance? It may be that users view these traits as more aligned with the functionalities of AI systems, so that capacities like knowledge and memory function as indicators of the reliability and accuracy of the systems. In contrast, the negative relationship between experience-related attributions and advice acceptance may stem from an interpretation of emotional AI as more volatile, biased, or unpredictable, leading to scepticism about its ability to provide accurate advice on factual tasks.

Despite common concerns about the consequences of consciousness attributions to AI for user trust, we found strong evidence against a positive association between consciousness attribution and advice-taking. This result may be in part due to the inherent complexity of consciousness attributions, which blend together intelligence-related traits (e.g., reasoning and memory) and experience-related traits (e.g., emotions and subjective awareness). In line with this, attributions of consciousness in the current data were related to both experience and (less strongly) intelligence. The null relationship between consciousness attributions and advice-taking may reflect the joint positive and negative effects of intelligence and experience. We note, however, that consciousness attributions in the current study were measured via a single question and relied on self-report. Implicit behavioural markers may reveal more nuanced attitudes.

Attributions of consciousness were lower in the current survey compared to data from the previous year using the same dependent measure (July 2023 $M = 25.56$; August 2024 $M = 21.51$)[6]. We speculate that this may be due to differences in the samples rather than usage decreasing attributions over time, given evidence that usage is associated with *greater* consciousness attributions[6] and exposure to ChatGPT increases attributions of mental states[7]. While the current results reflect a specific moment in time (since all participants completed the study in July and August 2024), future work may more directly explore longitudinal changes in attitudes, both in different samples and within the same participants, to explore how attitudes toward AI evolve over time.

Strikingly, attributions of consciousness to ChatGPT were specifically related to usage frequency of social AI, rather than ChatGPT itself or other AI systems such as digital assistants. This raises questions about how people reason about different kinds of AI systems, and how their experience with some platforms may generalize to others. While usage of different AI

systems may shape attributions of specific mental capacities, users' beliefs may also depend on characteristics of the users themselves, such as demographics or personality traits[27–29]. Future work may thus profitably examine how exposure to different systems interacts with user characteristics to shape beliefs about AI.

The current results also highlight an important dissociation between self-reported trust and participants' behaviour, since consciousness attributions were positively correlated with general impressions of trust but not with actual decisions in the advice-taking task. This is consistent with previous evidence that agent anthropomorphism has a greater effect on self-reported trust than actual trust behaviour, which is rather more sensitive to advice accuracy[30]. These effects highlight the importance of measuring trust not just via self-reports, but also via behavioural intentions as in the current study, as people may express trust verbally yet act with more caution, particularly when interacting with unfamiliar or non-human agents.

## Limitations
User attributions of mental states are but one determinant of trust, which is influenced by multiple factors including user characteristics, system characteristics, and interactive context[31]. For example, in the current study, advice-taking was operationalized as reversals of initial decisions in a general knowledge task. This task was chosen as it involves objective decisions where most people would find knowledge retrieval from ChatGPT helpful, while also involving some degree of high-level reasoning, thus leaving room for errors in knowledge synthesis and comparison. The relative influence of intelligence and experience attributions may, however, depend on the type of task. For example, the capacity for emotions may be more relevant than intelligence in advice-taking on more personal, emotion-involving matters such as relationships or mental health. Future work may thus examine the relative influence of metal state attributions on trust in various domains of decision-making.

Relatedly, the performance of ChatGPT in the current study was determined based on the accuracy of previous participants, and the advice was thus of comparable accuracy to the participants themselves[26]. This methodology allowed us to control the accuracy and format of the advice, but also resulted in advice that was sometimes incorrect—and indeed, participants accepted advice only on a subset of trials where they received discrepant advice (58%). Exposure to inaccurate advice may have influenced participants' willingness to trust the advice, as well as their beliefs about the system itself, impacting attributions of mental capacities like intelligence. An exploratory analysis showed that rates of advice-taking did not decrease significantly across the first and second half of the advice-taking task, suggesting limited learning. However, the current paradigm was not designed to detect such learning effects, and the advice-taking task was relatively short (with 40 trials in total), most of which featured advice that was in agreement with participants' responses. Importantly, attributions of intelligence and experience were similar in participants who provided their ratings before and after the advice-taking task, suggesting that exposure to incorrect choices did not significantly influence their beliefs about the system. Similarly, mental state attributions were unrelated to the overall accuracy of the advice received, even in the subset of participants who completed the ratings task following advice exposure. Nonetheless, effects of exposure to inaccurate advice may emerge in longer or repeated interactions, or in contexts where participants are penalized for providing inaccurate responses. Future work may explore high-stakes contexts or vary the accuracy of AI systems more systematically to investigate how stakes and perceived accuracy impact mental state attributions—and how prior beliefs about the capability of these systems regulate the way users update their estimates of the system's performance.

## Conclusions
Taken together, these findings highlight how human–AI interactions are shaped by complex and multifaceted inferences about the capacities of AI systems. In the conditions we examined, we found strong evidence against an overall positive relationship between consciousness attribution and advice-taking. Measurements of various mental state attributions revealed that attributions of intelligence-related traits enhanced trust, whereas experience-related traits tended to reduce trust.

These results open several avenues for future research probing specific characteristics of the AI system and the interaction context which may modulate these attributions and trust, both at the level of general impressions and at the level of actual behaviour. Further investigation will help the AI sector achieve well-calibrated and balanced trust, finding the middle ground between mistrust and over-reliance.

## Data availability
Anonymised raw data are openly available on the Open Science Framework (OSF) website at this link: https://osf.io/w537f/.

## Code availability
Analysis code is openly available on the OSF website at this link: https://osf.io/w537f/.

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

## Acknowledgements
The authors wish to thank Ali Boyle and Nick Shea for their feedback on experimental design. This work was supported by a UK Research and Innovation (UKRI) EPSRC Programme Grant (EP/V000748/1) and a Microsoft Research grant under the AI, Cognition, and the Economy (AICE) programme, and has benefitted from the Microsoft Accelerating Foundation Models Research (AFMR) grant programme. S.M.F. is a CIFAR Fellow in the Brain, Mind and Consciousness Programme, and is funded by UK Research and Innovation (UKRI) under the UK government's Horizon Europe funding guarantee [selected as ERC Consolidator, grant number 101043666 "ConsciousComputation"]. The funders had no role in study design, data collection and analysis, decision to publish or preparation of the manuscript.

## Author contributions
C.C. designed research, conducted the experiment, analysed data, and wrote the manuscript with input from J.B. and S.M.F.

## Competing interests
The authors declare no competing interests.
