## [Transparent Peer Review file · Communications Psychology]

The Influence of Mental State Attributions on Trust in Large Language Models

Corresponding Author: Dr Clara Colombatto

Version 0:

Decision Letter:

Dear Dr Colombatto,

Thank you for your patience during the peer-review process. I am sorry that the decision has been so much delayed. We now have received 3 reports on your manuscript "The Influence of Mental State Attributions on Trust in Large Language Models", which I include at the end of this message.

The reviewers find your work of interest but Reviewers #1 and #2 raised some important points. We are very interested in the possibility of publishing your study in Communications Psychology, but would like to consider your responses to the reports submitted by Reviewers #1 and #2 and assess a revised manuscript before we make a final decision on publication.

We therefore invite you to revise and resubmit your manuscript, along with a point-by-point response to the reviewers. Please highlight all changes in the manuscript text file.

Editorially, We ask that you address Reviewer #2's requests for further analyses and undertake textual revisions to improve the clarity of the description and discussion of limitations, as per both reviewers' comments. Editorially, we welcome the preregistration and use of Bayesian statistics, and ask you to ensure that your revision complies with our respective guidelines to facilitate future steps. For preregistrations, the key criterion is that all originally preregistered hypotheses and analyses are included, unless scientifically unsound, in which case the deviation needs to be highlighted and explained. Any additional analyses may be included, but need to be labelled as post-hoc, non-preregistered. The full policy is here: <https://www.nature.com/commpsychol/editorial-policies/preregistration-policy>

Regarding the interpretation of Bayesian statistics, we ask authors to avoid interpreting anecdotal evidence ($BF_{10} < 3$ and > 0.3), following the convention detailed in Schönbrodt, F.D., Wagenmakers, E. Bayes factor design analysis: Planning for compelling evidence. Psychon Bull Rev 25, 128–142 (2018) <https://rdcu.be/b6uOC>

I am attaching an Editorial Requests Table that details critical reporting requirements for the revised manuscript. Please attend to each item and ensure your manuscript is fully compliant. If your revised manuscript is not aligned with these requests on major issues, such as those concerning statistics, it may be returned to you for further revisions without re-review.

Please submit the following items:

- Revised manuscript
- Point-by-point response to the referees' comments
- Cover letter (as a separate document)
- <https://www.nature.com/documents/nr-reporting-summary.zip>>Nature Research Reporting Summary

- <https://www.nature.com/documents/nr-editorial-policy-checklist.pdf>>Editorial Policy Checklist
- Completed Editorial Request Table (attached).

via this link: Link Redacted .

Additional guidance is available in our style and formatting guide Communications Psychology formatting guide.

Best regards,

Marika, on behalf of

Erdem Pulcu
&
Jennifer Bellingtier

Marika Schiffer, PhD
Chief Editor
Communications Psychology

Dr Erdem Pulcu
Editorial board member
Communications Psychology
0000-0002-2170-0677

Dr Jennifer Bellingtier
Senior Editor
Communications Psychology

REVIEWER EXPERTISE:

Reviewer #1 cognitive psychology, decision making, LLMs
Reviewer #2 cognitive psychology, decision making, social cognition

REVIEWER REPORTS:

Reviewer #1 (Remarks to the Author):

The authors present interesting results on the relationship between trust and mental states that humans often attribute to LLMs. Specifically, they show that individuals who associate qualities related to intelligence with LLMs tend to trust them more in a decision-making task, while those who attribute emotional qualities to LLMs exhibit lower levels of trust in this task.

The paper offers an original contribution to the field. LLMs have been developed very fast and, to my knowledge, little has been done to understand how users comprehend them which is an important topic discussed in this paper.

Overall, the manuscript is well-written, clear, and easy to follow. However, I believe that the presentation of the advice-taking task could be slightly better framed. For instance, the phrase "advice from ChatGPT" (L 122) may be misleading, as the advice actually comes from previous participants, although participants are told it comes from ChatGPT. While this is clarified later in L 131, the initial wording could still cause confusion and could be reworded for greater clarity in order to avoid ambiguity.

Using advices from previous participants claiming they come from ChatGPT is an interesting methodological choice. It ensures the distribution of advices remains human-like which is important, however, this manipulation may influence participants' confidence in ChatGPT. For example, if a participant receives clearly incorrect advice - such as being told that Belgium has a larger population than India - it could severely undermine their trust in "ChatGPT". Since the study shows that people often perceive ChatGPT as intelligent, encountering errors labeled as ChatGPT's advice could bias participants' trust and behavior biasing the results. I don't think this potential methodological issue has been discussed in the paper.

In short, I believe the paper addresses an important and timely topic, shedding light on how users consider and interact with LLMs. The manuscript seems well written overall, but clarifying the wording of some methodological details and expanding the discussion a little to discuss more the limitations of the methods could further enhance its clarity in my opinion.

Reviewer #2 (Remarks to the Author):

Thank you for the opportunity to review the manuscript from Colombatto et al. Here, the authors ask participants to rate the mental states of an Open AI's ChatGPT, and then assess whether attributions of intelligence or consciousness were affiliated with advice-taking in a task of factual statements. Authors found that higher attributions of 'experience' - that is, items such as fear, guilt, and pleasure - were slightly related to lower advice-taking (although weakly), but attributions of intelligence - planning, knowing, and reasoning - were related to higher advice taking. The authors present a preregistered and open manuscript, with clear reporting of results, with some interesting findings that provide insight into how humans appraise artificial agents over varying degrees of familiarity (usage). I found the statistical tests clear, transparent and thorough. Findings also have implications for how humans may appraise other human qualities that increase advice-taking.

I have the following suggestions that I think could improve the manuscript:

Introduction

1. Line 31 - should be 'from' instead of 'front' I think
2. Your first sentence of the final paragraph is slightly vague, and I found myself a little lost as to what you were driving at. Which relationship between anthropomorphism and trust? which specific aspects may evoke more experiences of trust? I understand what you're driving at I think but I believe you could spell it out some more.

Methods

1. I am slightly concerned about the discrepancies in preregistered accounts and their reporting in the methods. The first is clear, although the second appears to need a little more justification. I understand that the first was adequately powered for the author's desired question, but given the lack of evidence, it was determined that more participants should be included until their H0 or H1 was meaningful in a Bayesian sense. I admire the transparency of the authors, however, when was the decision to recruit more based on inconclusive Bayesian outcomes determined? This caveat was not in the first preregistration and appears quite post hoc. I don't think this is necessarily a problem, but I think it needs to be explained a bit more, or I think the exploratory nature of the bayesian criteria used to justify the second recruitment drive should be highlighted a bit more.
2. It became obvious (but not before the section _Advice Taking Task_) that the ChatGPT API was not exposed to participants but instead participants were matched to other participant responses. The cover story of using ChatGPT (but not exposing the API) needs to be clearer earlier to make it obvious that this is a psychological manipulation rather than an actual interaction with ChatGPT.
3. You paired participants with other real human answers, but was there any matching based on the correctness of the previous participants? That is, was it entirely random whether a participant with 70% or 30% accuracy was given to participants? Did you control for this in the way that participants responded or learned about their partner?

Results

1. Was there any learning curve concerning how likely participants were to accept advice with mostly correct/incorrect partner pairs?
2. Small point: The distribution of advice taken seems to have a reasonable proportion of people at the '0' end - was this at all related to the correctness of their partner? Did this floor influence the results in any sense?
3. The point about the usage of mental state attributions and usage is very interesting, and I wonder if it perhaps reveals more about the user than anything else. Do the authors have any data on the profiles of the participants (e.g. mental state attributes of themselves) and how they relate to the attributed profiles of their 'ChatGPT' partner?

Discussion

Did you manage to capture the time when this data was collected? I noticed your preregistration were mid last year. I ask because the popular useage of AI chatbots were at its zenith around last year (before brick walls and performance issues

were clearly noted in the newer models) and I wonder to what degree you speculate that the hype around AI may have played into participant's perceptions of trust and safety in the AI they were 'playing' with?

I look forward to reading the author response, and thank you once again for the opportunity to review this work.

Reviewer #3 (Remarks to the Author):

This manuscript makes a significant and original contribution to our understanding of how mental state attributions influence trust in artificial intelligence systems. The study's key strength lies in its methodologically rigorous examination of how different dimensions of mental state attribution - intelligence versus experience - distinctly affect user trust and behavior. The research is technically sound, with appropriate preregistered analyses, robust sample size (N=410), and comprehensive statistical reporting using both frequentist and Bayesian approaches. The findings challenge existing assumptions about anthropomorphism and trust, revealing that while intelligence-related attributions positively correlate with trust, experience-related attributions show a negative relationship. The study was conducted ethically with proper oversight, and the authors have made their data and analysis code openly available. The results have important implications for both theoretical understanding of human-AI interaction and practical applications in AI system design. The paper presents clear evidence for its conclusions and addresses an important question relevant to multiple psychological sub-fields. Given the methodological rigor, theoretical contribution, and practical significance, I recommend acceptance without revision.

Version 1:

Decision Letter:

Dear Dr Colombatto,

Your manuscript titled "The Influence of Mental State Attributions on Trust in Large Language Models" has now been seen by our reviewers, whose comments appear below. In light of their advice I am delighted to say that we are happy, in principle, to publish a suitably revised version in Communications Psychology.

We therefore invite you to revise your paper one last time to address the remaining concerns of our reviewers and a list of editorial requests. At the same time we ask that you edit your manuscript to comply with our format requirements and to maximise the accessibility and therefore the impact of your work.

EDITORIAL REQUESTS:

SUBMISSION INFORMATION:

OPEN ACCESS:

* DATA AVAILABILITY:

Link Redacted

Best regards,

Jennifer Bellingtier

Jennifer Bellingtier, PhD
Senior Editor
Communications Psychology

Dr Erdem Pulcu
Editorial board member
Communications Psychology
0000-0002-2170-0677

REVIEWERS' EXPERTISE:

Reviewer #1 cognitive psychology, decision making, LLMs
Reviewer #2 cognitive psychology, decision making, social cognition

REVIEWERS' COMMENTS:

Reviewer #1 (Remarks to the Author):

The authors addressed my concerns very well. I recommend acceptance.

Reviewer #2 (Remarks to the Author):

The authors have thoroughly and satisfactorily addressed all of my comments. I look forward to seeing it in press. It is now ready for publication.
